# Charged Residue Implantation Improves the Affinity of a Cross-Reactive Dengue Virus Antibody

**DOI:** 10.3390/ijms23084197

**Published:** 2022-04-11

**Authors:** Huiling Wei, Jie Tan, Bingjie Zhou, Xiaotong Guan, Qiaoxian Zhong, Jiaqi Wang

**Affiliations:** School of Pharmaceutical Sciences (Shenzhen), Sun Yat-Sen University, Shenzhen 518107, China; weihling@mail2.sysu.edu.cn (H.W.); tanj67@mail2.sysu.edu.cn (J.T.); zhoubj9@mail2.sysu.edu.cn (B.Z.); guanxiaotong@nibs.ac.cn (X.G.); zhongqx3@mail2.sysu.edu.cn (Q.Z.)

**Keywords:** cross-reactive antibody, dengue virus, affinity maturation, charge complementarity, molecular dynamics simulation

## Abstract

Dengue virus (DENV) has four serotypes that complicate vaccine development. Envelope protein domain III (EDIII) of DENV is a promising target for therapeutic antibody development. One EDIII-specific antibody, dubbed 1A1D-2, cross-reacts with DENV 1, 2, and 3 but not 4. To improve the affinity of 1A1D-2, in this study, we analyzed the previously solved structure of 1A1D-2-DENV2 EDIII complex. Mutations were designed, including A54E and Y105R in the heavy chain, with charges complementary to the epitope. Molecular dynamics simulation was then used to validate the formation of predicted salt bridges. Interestingly, a surface plasmon resonance experiment showed that both mutations increased affinities of 1A1D-2 toward EDIII of DENV1, 2, and 3 regardless of their sequence variation. Results also revealed that A54E improved affinities through both a faster association and slower dissociation, whereas Y105R improved affinities through a slower dissociation. Further simulation suggested that the same mutants interacted with different residues in different serotypes. Remarkably, combination of the two mutations additively improved 1A1D-2 affinity by 8, 36, and 13-fold toward DENV1, 2, and 3, respectively. In summary, this study demonstrated the utility of tweaking antibody-antigen charge complementarity for affinity maturation and emphasized the complexity of improving antibody affinity toward multiple antigens.

## 1. Introduction

Dengue is a tropical and subtropical disease, and due to climate change, it has spread to a broader area [1]. The dengue virus (DENV) belongs to the flavivirus family and has four serotypes. Secondary infection with a different serotype might cause severe dengue symptoms. Antibody-dependent enhancement (ADE) has been recognized as a potential mechanism responsible for severe dengue. Previous studies showed that non-neutralizing antibodies or sub-neutralizing concentrations of neutralizing antibodies can cause ADE in vitro and in vivo [2]. Therefore, an ideal therapeutic antibody must be able to neutralize all serotypes with comparable potencies to minimize the risk of ADE.

Our group and others have developed broadly neutralizing bispecific antibodies against a closely related flavivirus—Zika virus [3], and different DENV serotypes [4]. As an alternative, broadly neutralizing antibodies are highly desirable for antiviral therapeutic development but rarely emerge in natural immune responses. Human humoral responses to DENV infection were shown to be dominated by antibodies to pre-membrane protein and the fusion loop in the envelope protein [5]. Recent studies have found serotype-specific neutralizing antibodies bound complex, quaternary envelope protein epitopes on the virus surface, especially in the hinge region connecting envelope protein domain I and II [6,7,8]. In contrast, broadly neutralizing antibodies recognized the envelope protein dimer epitope [9]. In addition, envelope protein domain III (EDIII)-specific antibodies constituted a minor component of the human humoral response but have high potency [10]. Antibodies targeting DENV EDIII include serotype-specific antibodies binding to the FG loop [11], poorly-neutralizing cross-reactive antibodies targeting the AB loop [12], or cross-reactive antibodies targeting A/G-strand [13,14]. One cross-reactive neutralizing antibody named 1A1D-2 binds DENV1, 2, and 3 but not 4 [13]. As EDIII is not an immunodominant epitope, therapeutic usages of anti-EDIII antibodies do not risk competing with naturally occurring neutralizing antibodies. Therefore, antibodies targeting EDIII serve as promising candidates for immunotherapy development. However, anti-EDIII cross-reactive antibodies generally have low affinities and require further affinity maturation to improve neutralizing potencies against all DENV serotypes.

Traditional methods for antibody engineering include phage and yeast surface display screening, which are costly and lengthy processes. Alternatively, structure-guided rational design requires an antigen-antibody complex structure, and considerable successes have been achieved [14,15]. However, antibody affinity improvement toward multiple antigens is still challenging due to sequence variations of epitopes. Furthermore, few studies have investigated the mechanism of affinity enhancement toward different antigens.

In the present study, the interface between 1A1D-2 and DENV2 EDIII was analyzed to discover unsatisfied charged residues in the epitope based on the previously solved crystal structure (PDB code 2R29). Mutations of 1A1D-2 were then designed and further validated using molecular dynamics (MD) simulation showing that the mutations can potentially form new electrostatic interactions with the epitope. Subsequently, binding kinetics were measured for these mutants toward recombinant EDIII of different serotypes. Additional MD simulations were used to investigate molecular mechanisms of affinity improvement toward different serotypes.

## 2. Results

### 2.1. Structural Analysis

The crystal structure of 1A1D-2 and DENV2 EDIII has been determined using X-ray crystallography to 3 Å [13]. The 1A1D-2 epitope on DENV2 EDIII involved A-strand (305–312), BC loop (323, 325, 327), DE loop (361, 362, 364), G-strand (385–391, 393). The paratope includes heavy chain residues in HCDR1 (26–28, 30–33), HCDR2 (52, 54–55), HCDR3 (98–102, 104–105) and light chain residues in LCDR1 (32–34, 36), LCDR2 (50, 53–54, 57–60). Out of the 22 residues in the epitope, ten of them have either positively or negatively charged side chains, including Lys305, Lys307, Lys310, Glu311, Arg323, Glu327, Lys361, Asp362, Lys388, and Lys393 (Figure 1A). The antibody-antigen interface is complementary in shape, with a negatively charged groove in the antibody binding to the protruded and positively charged A-strand (including Lys305, Lys307 and Lys310) of EDIII (Figure 1A). However, among the ten charged residues in the epitope, side chains of seven were not involved in hydrogen bonds or electrostatic interactions with the paratope, including Glu311, Arg323, Glu327, Lys361, Asp362, Lys388, and Lys393.

### 2.2. Mutation Design to Improve Affinity

To improve affinity, we designed mutations of 1A1D-2 in order to form new electrostatic interactions with unsatisfied charged residues in the epitope mentioned above. One of the epitope residues, Arg323, did not form any direct hydrogen bond or salt bridge with the antibody according to the crystal structure. Atoms in 1A1D-2 near Arg323 include the CB of Ala54.H (4.6 Å), O atoms of Lys30.H (5.3 Å) and Asp31.H (5.4 Å), and OD1 of Asn55.H (5.9 Å). Among them, the Ala54.H side chain pointed toward Arg323 (Figure 1B). In order to improve affinity, we suggested that a mutation of Ala54.H to the negatively charged glutamic acid (A54E.H) can potentially form a salt bridge with Arg323, therefore strengthening the binding.

For another residue, Glu327, its side chain is near Glu59.L of 1A1D-2, which might exert repulsive forces to each other (Figure 1C). However, close examination of residue Glu59.L revealed that its side chain formed hydrogen bonds with the N of Ser60.L (1A1D-2) and the NZ atom of Lys307 (DENV2 EDIII). Therefore, Glu59.L might be important for binding and was kept to preserve the interaction. Further check of residues around Glu327 in 1A1D-2 suggested that substituting Tyr105.H with arginine (Y105R.H) might introduce extra electrostatic interactions to Glu327 (Figure 1C).

In summary, Y105R.H and A54E.H were predicted to form new electrostatic interactions with the antigen (DENV2 EDIII) for affinity improvement. In this study, we tested these two mutations as a proof of concept for affinity maturation by introducing charged residues to the paratope.

### 2.3. MD Simulation Validates the Feasibility of the Designs

MD simulation was conducted to validate the feasibility of the design. The mutated antibody-DENV2 EDIII complexes were constructed based on the 1A1D-2-DENV2 EDIII crystal structure. DENV2 EDIII in complexes with 1A1D-2 variable fragment (Fv) (WT and mutants) were then energy minimized and pre-equilibrated as described in the Methods section (Section 4.4). MD production runs were conducted for each complex for 500-ns. All three complexes remained stable with root mean square deviation (RMSD) values of all backbone atoms at 2–3 Å (Appendix A). To validate whether Y105R.H and A54E.H formed the predicted salt bridges, inter-residue interactions were analyzed based on the last 50-ns trajectories of the simulation using the gmx_MMPBSA tool [18].

The results showed that Glu54.H has not only formed the predicted salt bridge with Arg323, but also interacted with Lys310 (Figure 2A). The averaged distance between OE1 of Glu54.H to NH1 of Arg323 was 3.3 ± 0.6 Å for the last 50-ns of MD simulation, indicating formation of a stable salt bridge (Figure 2B,C). Taking 4 Å as the threshold for the salt bridge formation, the Glu54.H-Arg323 salt bridge was maintained 90% of the time in the last 50-ns. In comparison, the interaction between Glu54.H and Lys310 was less stable with a distance of 4.3 ± 1.1 Å between OE1 of Glu54.H and NZ of Lys310 (Figure 2B,C). Consistently, decomposition of the binding energy for Glu54.H revealed that Arg323 made more contribution than Lys310 to the binding (Figure 2D). This analysis also revealed that the mutated residue itself (e.g., Glu54.H) has a positive value in binding energy contribution, probably due to a high desolvation energy price (Figure 2D).

Consistent with the crystal structure, Glu327 in DENV2 EDIII did not form direct hydrogen bond or ionic pair with 1A1D-2 WT in the MD simulation. In contrast, new salt bridges were established between Arg105.H and Glu327 in the Y105R.H-DENV2 EDIII complex (Figure 3A,B). Surprisingly, Arg105.H also made an electrostatic contact with Asp362 (Figure 3A,B). During the simulation, the electrostatic attractions were alternating between Glu327 and Asp362 for Arg105.H (Figure 3A). The averaged distance was 5.0 ± 1.4 Å between NH2 of Arg105.H and OE2 of Glu327, and 5.1 ± 2.9 Å between NH2 of Arg105.H and OD1 of Asp362 (Figure 3D). Decomposition of the binding energy for Arg105.H revealed that both Glu327 and Asp362 made similar contributions (Figure 3E).

According to the simulation results, we concluded that these mutations have a high chance to form the predicted salt bridges with the antigen. Therefore, they were expressed and purified, and their affinities with recombinant EDIII were measured.

### 2.4. Binding Kinetics

Surface plasmon resonance (SPR) experiments were conducted with purified Fab of 1A1D-2 WT and mutants against immobilized recombinant EDIIIs. As expected, both A54E.H and Y105R.H improved 1A1D-2 affinity to recombinant EDIII of DENV2 (Figure 4, Table 1). The affinity of 1A1D-2 WT, A54.H, and Y105R.H toward recombinant EDIII of DENV2 were 440, 41.9, and 203 nM, respectively. The affinity improvement for A54.H and Y105R.H toward DENV2 EDIII were 10.5- and 2.2-fold, respectively. SPR experiments also showed that the two mutations improved affinity of 1A1D-2 toward recombinant EDIII of DENV1 and 3.

The affinity of WT 1A1D-2 toward EDIII of DENV1 and DENV3 are 216 and 91 nM, respectively (Figure 4, Table 1). For DENV1, A54E and Y105R improved affinities by 1.5- and 5.1-fold compared to WT, respectively. For DENV3, A54E.H, and Y105R.H improved affinities by 8.8- and 1.8-fold, respectively. The trends of affinity improvement by A54.H and Y105R.H were similar toward DENV2 and DENV3. Previous study has shown that 1A1D-2 is unable to neutralize DENV4. Consistently, our result showed that WT did not bind the recombinant DENV4 EDIII. In addition, neither Y105R.H or A54E.H bound DENV4 EDIII in our SPR experiments.

SPR experiments also revealed the binding kinetics for the interactions. For 1A1D-2 WT, DENV2 has a much faster dissociation rate (4.71 × 10^−2^/s) compared to DENV1 (7.55 × 10^−3^/s) and DENV3 (7.92 × 10^−3^/s), indicating a longer half-life of binding to DENV1 and 3. The association rate of WT to DENV1, 2, and 3 are 3.49 × 10^4^, 1.07 × 10^5^, and 8.7 × 10^4^/Ms, respectively. The affinity improvement by A54E.H to DENV2 and DENV3 is due to faster association rates and slower dissociation. For DENV2, k_a_ increased by 2-fold and k_d_ decreased by approximately 6-fold. For DENV3, k_a_ increased by 4-fold and k_d_ decreased by approximately 2-fold. For mutant Y105R.H, the affinity improvement is mainly due to a slower dissociation rate compared to WT (Table 1).

### 2.5. Double Mutant Additively Improves 1A1D-2 Affinity

As A54E.H and Y105R.H are located distantly to each other and interact with different residues in the epitope, we suggested that combination of these two mutations could further improve affinity. As expected, the double mutant containing Glu54.H and Arg105.H showed an additive effect. SPR results showed that A54E-Y105R double mutant bound EDIIIs of DENV1, 2 and 3 at an affinity of 27.4, 12.3, and 7.1 nM, respectively, which corresponded to 8-, 36-, and 13-fold increase in affinity compared to the WT (Figure 5, Table 1). As expected, the double mutant did not bind DENV4 EDIII.

### 2.6. Molecular Mechanism of Affinity Improvement to DENV1 and DENV3

Sequences alignment showed that residues interacting with mutated residues of 1A1D-2 were not fully conserved in DENV1 and DENV3 (Figure 6). Residue 327 (325 in DENV3) in DENV1 and 2 are both glutamic acid, and replaced by lysine in DENV3. However, residues 362 (360 in DENV3) is negatively charged Glu, Asp and Glu in DENV1, 2 and 3, respectively. Residue 323 (equivalent to 321 in DENV3) is Gln, Arg, and Lys in DENV1, 2, and 3, respectively. Only Lys310 (equivalent to 308 in DENV3) is conserved among all four serotypes. These sequence variations have affected the extent of affinity improvement for the two mutants toward different serotypes.

To probe potential new interactions of the two mutations toward DENV1 and DENV3, complexes of 1A1D-2 Fv (WT, Y105R.H, A54E.H)-EDIII (DENV1, 3) were modeled. The EDIII structures of DENV1 and 3 were extracted from PDB 3UZQ and 3VTT. The EDIII of the corresponding serotype was superimposed to DENV2 EDIII in the structure of 1A1D-2 Fab-DENV2 EDIII complex (PDB code 2R29). All complexes were stable in the 500-ns MD simulation with similar RMSD values at approximately 2–3 Å for all backbone atoms (Appendix A).

#### 2.6.1. Affinity Improvement of A54E.H to DENV1 and 3

The mutations were initially designed based on the DENV2 EDIII structure. For DENV2, A54E mutation of 1A1D-2 formed salt bridges with Arg323 and Lys310 (Figure 2). As mentioned above, the affinity of A54E.H to DENV1, 2, and 3 improved 1.5-, 10.5-, and 8.8-fold, respectively (Table 1, Figure 4). In the A54E.H-DENV3 complex, A54E.H mainly formed new salt bridges with Lys308 (equivalent to 310 in other serotypes), and weakly interacted with Lys321 (equivalent to 323 in other serotypes) (Figure 7A). The weaker interaction between Glu54.H and Lys321 in DENV3 could be due to the shorter side chain of Lys321 in DENV1 compared to Arg323 in DENV2 (Figure 7C,D). Different from DENV2 and DENV3, residue 323 in DENV1 EDIII is the neutral glutamine, which showed no apparent interaction with Glu54.H. In our simulation, Glu54.H only occasionally interacted with Lys310 in DENV1 and the interaction was quite unstable for the whole trajectory (Figure 7B,E). This potentially explains the lower affinity improvement of A54E.H to DENV1 EDIII compared to other serotypes.

#### 2.6.2. Affinity Improvement of Y105R.H to DENV1 and 3

For DENV2, Y105R mutation of 1A1D-2 formed salt bridges with Glu327 and Glu362 (Figure 3). As residue 325 (equivalent to 327 in other serotypes) in DENV3 is a lysine, it might introduce repulsive forces to Arg105.H. However, experimental results showed a 1.8-fold affinity improvement for Y105R.H to DENV3 compared to WT (Figure 4, Table 1). Therefore, we proposed that Y105R.H should mainly contact a different set of residues in DENV3. For the Y105R.H-DENV3 complex, analysis of the trajectory revealed that Arg105.H still contacted Glu360 (equivalent to 362 in other serotypes). In addition, Arg105.H also strongly interacted with Glu323 (equivalent to 325 in other serotypes) (Figure 8A). Glu323 is two-residue upstream of Lys325 in DENV3 (327 in other serotypes) (Figure 6). We also noticed that Lys307 in DENV3 has repulsive forces on Arg105.H, indicated by a positive binding free energy (Figure 8A,D). Decomposition of the binding energy for Arg105.H to DENV3 showed that Glu323 made more contribution than Glu360 to the binding (Figure 8D). Surprisingly, Lys325 in DENV3 did not introduce repulsive forces to Arg105.H. Energy decomposition revealed that it interacted with Tyr53 and Glu59 in the light chain (Appendix A). This interaction might pull Lys325 away from Arg105.H, diminishing its chance to unfavorably contact Arg105.H.

Y105R.H has a higher affinity improvement toward DENV1 (5.1-fold) compared to DENV2 (2.2-fold) and DENV3 (1.8-fold). Surprisingly, analysis of the trajectory showed that Arg105.H did not strongly interact with DENV1 during the last 50-ns of our MD simulation. However, Arg105.H did strongly interact with DENV1 EDIII in some episodes within the 500-ns simulation (e.g., 200 to 275-ns) (Appendix A). This could be limited by insufficient samplings of the current MD simulation, for which the Y105R.H-DENV1 EDIII complex might have not reached the most stable conformation. The interface was analyzed based on the trajectory from 200 to 250-ns, and the result showed that Arg105.H mainly formed a salt bridge with Glu362 but not Glu327 (Figure 8B,E). Therefore, affinity improvement by Y105R.H mutant was achieved by making subtly different molecular contacts.

## 3. Discussion

Broadly neutralizing antibodies against multiple virus strains, genotypes, or serotypes are highly desirable. However, affinity improvement of an antibody to multiple targets is challenging. Identities of EDIII sequences are around 60% among four DENV serotypes. In this study, two mutations designed by introducing charge complementarity were shown to improve affinities of a cross-reactive antibody toward DENV1, 2, and 3. It would require neutralization experiments to validate whether these affinity-improved mutations of 1A1D-2 will improve its potency against DENV1, 2, and 3.

Molecular mechanisms of antibody affinity maturation inspire structure-based rational design. One previous study showed that rigidification of the initially flexible HCDR3 to its bound conformation could account for most of the affinity gain for antibodies [19,20]. Alternatively, mutations in the heavy chain-light chain (HC/LC) interface improved antibody affinity by altering their relative orientation to conformations competent to bind antigens [21]. Furthermore, protein engineering based on simplified energy functions showed that introducing hydrogen bonds or electrostatic interactions efficiently improves affinity [22]. In this study, we showed that mutations with charge complementarity to the epitope could improve antibody affinity.

Recent studies have attempted to improve the affinity of cross-reactive antibodies. One group has engineered the DENV antibody 4E11 to improve its affinity toward DENV4 [14,23]. 4E11 neutralizes DENV1, 2, and 3 but not DENV4 due to its poor affinity to this serotype. An empirical approach that captures physicochemical features common in antigen-antibody interfaces was applied to predict affinity-improved mutations [14]. Consistent with our finding, successful mutations were mostly charged or polar and resided at the periphery of the binding interface [14]. Our study further revealed that the improved affinity to each antigen could be due to specific interactions with different epitope residues. This study is also in line with previous findings suggesting that the flexibility of design sequences is essential for affinity improvement to one antigen without compromising the other [15,24].

It will require further mutations to make 1A1D-2 fully cross-reactive toward all serotypes. Engineering of 1A1D-2 to bind DENV4 demands a clear understanding of the molecular basis for its incompetence to bind DENV4. Since EDIII of DENV4 has a similar structure to that of other serotypes, we speculate that the inability of binding could be due to either steric clashes or repulsive forces between 1A1D-2 and key residues in the epitope site of DENV4. If this is the case, it is likely that antibody engineering to make 1A1D-2 bind DENV4 would weaken its binding to other serotypes. However, considering the flexibility of the epitope surface illustrated in this study, we think that further mutations could be designed to compensate for the potential affinity loss to other serotypes.

In this study, two mutants presented different profiles of binding kinetics improvement. The increased association rate by A54E.H suggested that Glu54.H might strengthen the attractive force of 1A1D-2 to antigens. Consistently, Selzer et. al. has hypothesized long-range electrostatic interactions increased protein-protein association rates [25]. In addition, the different binding kinetics changes for A54E.H and Y105R.H could be due to exact chemical environments around the mutations. For DENV2, the introduced Arg105.H is located not only close to negatively charged residues (e.g., Glu327 and Asp362) but also positively charged ones (e.g., Lys307 and Lys361) in EDIII (Figure 1). The overall electrostatic environment might not be optimal for this mutation. Nevertheless, it can stabilize the complex once the salt bridge is formed. In contrast, A54E.H is located at the peripheral of the interface, with both Arg323 and Lys310 in DENV2 EDIII attractive to Glu54.H (Figure 1B).

Previous studies have suggested that DIII-binding antibodies constituted a minor component of the human humoral response against DENV but have high potency [10]. Wahala et al. showed that cross-reactive EDIII-binding antibodies arose after secondary infection of DENV [26]. However, it is shown that naturally occurring EDIII-binding human antibodies are poorly neutralizing, albeit with full cross-reactivity to all four serotypes [10]. Antibody-antigen complex structures have been solved for several cross-reactive EDIII-binding antibodies [12,13,27]. Some of these antibodies target the AB-loop epitope, which is buried in the E protein dimerization interface and poorly exposed on the virus surface. Antibodies targeting this epitope are weakly or non-neutralizing [12,28,29]. Another class of antibodies, including 1A1D-2 and 4E11, target the A/G-strand epitope, which is less conserved compared to the AB-loop [13,27]. Structural studies have shown that the A/G-strand epitope is already partially exposed in the compact mature virus structure, and becomes fully exposed at the expanded virus structures [13,30,31]. A germline-like human antibody, m366.6, is also fully cross-reactive and suggested to target a similar epitope as 1A1D-2 although with a different bound orientation [32]. Among all these cross-reactive EDIII-binding antibodies, those targeting A/G-strand seem more promising to be developed as potential therapeutics. Our study has contributed to insights for antibody affinity maturation toward multiple antigens by purposely optimizing electrostatic complementarity of the interface.

In conclusion, A54E.H and Y105R.H improved 1A1D-2 affinity to EDIIIs of DENV1, 2, and 3 by different molecular contacts. Affinity maturation could be effectively achieved by residue substitution in the antibody optimizing charge complementarity to the epitope. This study highlighted the flexibility of the DENV EDIII epitope to accommodate charged residue substitution and the complexity to engineer a cross-reactive antibody against multiple antigens. Further antibody engineering could be attempted to enable 1A1D-2 to bind DENV4 in achieving a full cross-reactivity.

## 4. Materials and Methods

### 4.1. Cloning, Expression, and Purification of Four Recombinant EDIIIs

DENV1-4 EDIII sequences were synthesized in Sangon Biotech, corresponding to protein sequences in DENV1 strain GuianaFGA891989, DENV2 strain Thailand/16681/84, DENV3 strain Philippines/H87/1956, DENV4 strain Myanmar1976. EDIII genes were amplified with primer pairs (Appendix A) and cloned into pMAL-c2x vectors (Miaolingbio, Wuhan, China) with SacI and XhoI as restriction sites. EDIIIs were expressed as N-terminal MBP fusion proteins in E. coli BL21(DE3) cells. Protein expression was induced with 0.5 mM Isopropyl-β-D-thiogalactopyranoside (IPTG) at 37 °C for 4 h. Proteins were extracted by sonication in ice-cold buffer (20 mM Tris-HCl, 200 mM NaCl, 1 mM EDTA, pH 7.4), and purified via MBPTrapTM HP column as recommended (GE Healthcare, Piscataway, NJ, USA). The proteins were concentrated and buffer-exchanged to PBS buffer (1.47 mM KH_2_PO_4_, 2.7 mM KCl, 4.3 mM Na_2_HPO_4_, 137 mM NaCl, pH 7.4).

### 4.2. Cloning and Preparation of 1A1D-2 Fragment Antigen Binding (Fab)

Gene of 1A1D-2 Fab (PDB 2R29) was synthesized and cloned into pComb3XSS (Miaolingbio, Wuhan, China). A54E.H and Y105R.H mutants were obtained by site-directed mutagenesis (Appendix A). Antibody expression was done using E.coli Top10F’ competent cells (ToloBio, Shanghai, China). In short, cultures were grown with 2 × Yeast Extract Tryptone medium supplemented with 100 μg/mL of ampicillin, 50 μg/mL of tetracycline and 20 mM of MgCl_2_ at 37 °C at 250 rpm until OD600 reached 0.6–0.8. Antibody expression was induced by 1 mM of IPTG, overnight at 28 °C at 250 rpm. Periplasmic fractions were harvested by an osmotic shock method as previously described [33]. Samples were further purified using a HiTrapTM Protein L column (Cytiva) as recommended.Purified Fabs were concentrated and exchanged into PBS buffer.

### 4.3. Binding Kinetics by Surface Plasmon Resonance (SPR)

Binding kinetics of 1A1D-2 Fab to EDIII were analyzed by SPR (Biacore T200, GE Healthcare, Uppsala, Sweden). Antibodies and antigens were first polished using Superdex 200 Increase 10/300GL column in PBS buffer. Purity of proteins was confirmed via 10% sodium dodecyl sulfate polyacrylamide gel electrophoresis. Recombinant MBP-EDIIIs diluted in 10 mM of sodium acetate buffer (pH 4.5) were captured on CM5 sensor chips (GE Healthcare, Uppsala, Sweden) via amine coupling as recommended. Antibodies diluted in a two-fold series in a range between 0.78–3200 nM were flowed through the chip at 30 μL/min for 120 s. Bound antibodies were allowed to dissociate for 180 s (or 480 s for double mutant) with flowing of PBST buffer (PBS plus 0.05% Tween) before the surface was regenerated using 10 mM of glycine buffer (pH 2.5). Binding kinetics were determined using Biacore T200 Evaluation software version 3.1 (GE Healthcare, Uppsala, Sweden).

### 4.4. MD Simulation

DENV1 and DENV3 EDIIIs were from PDB 3UZQ and 3VTT, respectively. Structures of 1A1D-2 variable fragment (Fv) in complex with DENV1 and DENV3 EDIII were generated by superimposing EDIIIs to the 1A1D-2-DENV2 EDIII structure (PDB 2R29). Antibody mutants were generated in Pymol [17] with rotamers selected based on minimal clashes. The topology and coordinate for complexes were generated using the pdb2gmx module in the GROMACS suite [34] using the Amber99SB-ILDN force field [35].

Each complex was placed in a dodecahedron box with the periodic boundary condition buffered for 1 nm at each dimension. The system was solvated using the TIP3P solvent model and neutralized with an ionic strength of 0.15 M. Complexes were minimized to release clashes, and NPT equilibrations were conducted at 298 K and 1 atm for 5-ns. The production runs were conducted for 300-ns with a 2-fs step. Bonds involving H are constrained using parallel LINCS methods. Long-range interactions were calculated using the Particle-Mesh-Ewald summation method. The simulation trajectory was post-processed using the trjconv module, visualized and analyzed in Pymol [17]. The RMSD and atom distance were calculated by GMX rms and distance modules, respectively.

### 4.5. In Silico Analysis of Binding Mechanisms

The binding was analyzed using the gmx_MMPBSA tool based on the GROMACS MD trajectories [18]. The binding free energy was computed using the molecular mechanics energies combined with generalized Born and surface area continuum solvation (MM/GBSA) method, with GB-OBC1 model for GB [36]. Snapshots were extracted from the last 50-ns of the 500-ns MD simulation with an interval of 2.5-ns for analysis of most complexes. For the Y105R-DENV1 EDIII complex, the trajectory from 200 to 250-ns was used for snapshot extraction. The binding energy decomposition was conducted using “idecomp 4” scheme described in http://valdes-tresanco-ms-github.io/gmc_MMPBSA/input_file/#the-input-file, accessed on 20 February 2022. Relevant figures were prepared by the gmx_MMPBSA_ana tools.

## Figures and Tables

**Figure 1 ijms-23-04197-f001:**
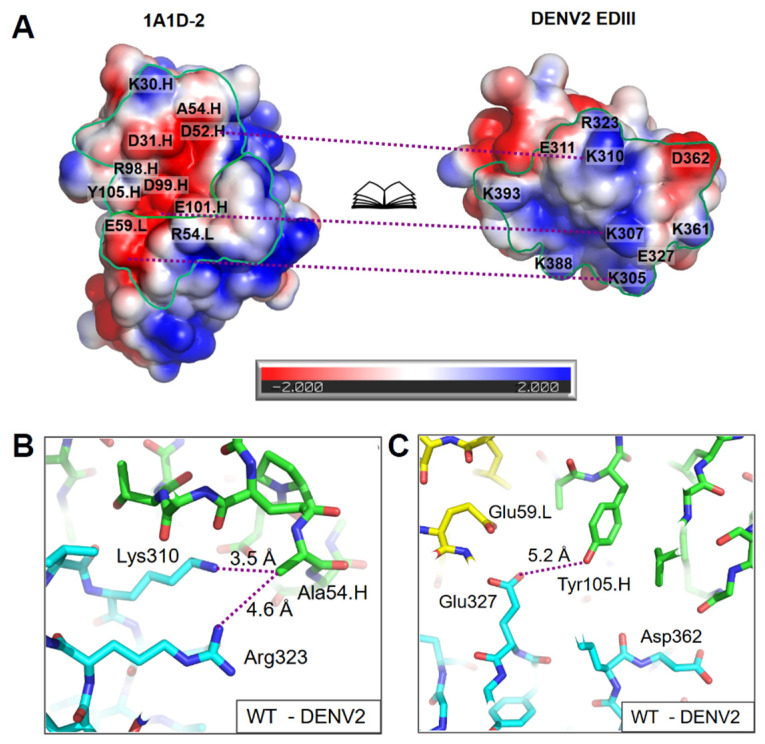
Design of antibody mutation. (**A**) Open-book view of the electrostatic surface for the binding interface between 1A1D-2 and DENV2 EDIII, calculated by APBS [16]. The potential at the solvent accessible surface is colored in blue (positive charge), red (negative charge), and white (neutral), with the coloring bar at the bottom. The figure was prepared using Pymol [17]. Paratope and epitope are highlighted with green lines. The purple dashed lines connect areas which interact with each other. Based on the crystal structure of 1A1D-2 antibody binding fragment (Fab)-DENV2 EDIII complex, (**B**) the distances between CB of A54 in the heavy chain and NH1 of R323 and NZ of K310 are labeled, respectively; (**C**) the distance between OH of Y105 in the heavy chain and OE1 of Glu327 is labeled. Carbon atoms in EDIII, antibody heavy chain, and light chain are colored in cyan, green, and yellow, respectively. Nitrogen and oxygen atoms are colored in blue and red. Purple dashed lines in (**B**,**C**) connect the atom pairs between which the distances are measured.

**Figure 2 ijms-23-04197-f002:**
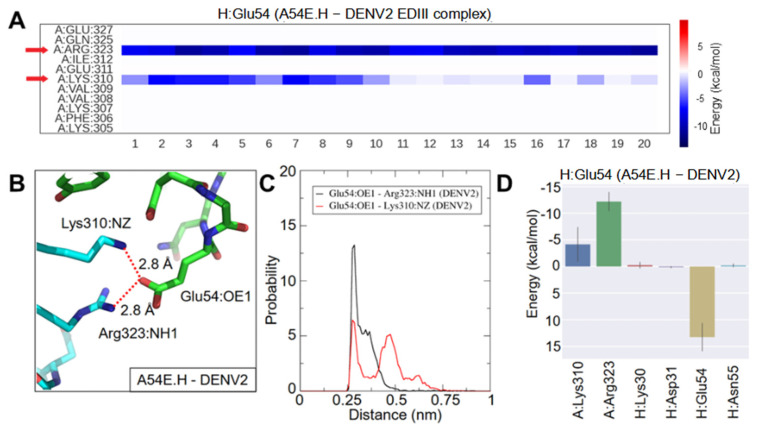
MD simulation of A54E.H with DENV2 EDIII. (**A**) Energy decomposition analysis: heat map shows energy contribution of residues that interact with Glu54.H per frame in the A54E.H-DENV2 EDIII complex. Twenty frames with a 2.5-ns interval were extracted for the last 50-ns of the MD simulation. Red arrows highlight residues strongly interacting with Glu54.H. (**B**) New salt bridges were formed between (1) OE1 of Glu54.H and NH1 of Arg323 and (2) OE1 of Glu54.H and NZ of Lys310 in the A54E.H-DENV2 EDIII complex. The interacting atoms are connected by red dashed lines with distances labeled in a representative snapshot. (**C**) Histograms of atom pair distances in the last 50-ns trajectory. Black represents the distance between OE1 of Glu54 and NH1 of Arg323, and red represents the distance between OE1 of Glu54 and NZ of Lys310. (**D**) Energy contribution of residues that interacted with Glu54.H, corresponding to the heatmap in (**A**). The bars represent averages and solid lines represent the standard deviation. The EDIII, 1A1D-2 heavy chain, and light chain are designated as A, H, and L chain, respectively.

**Figure 3 ijms-23-04197-f003:**
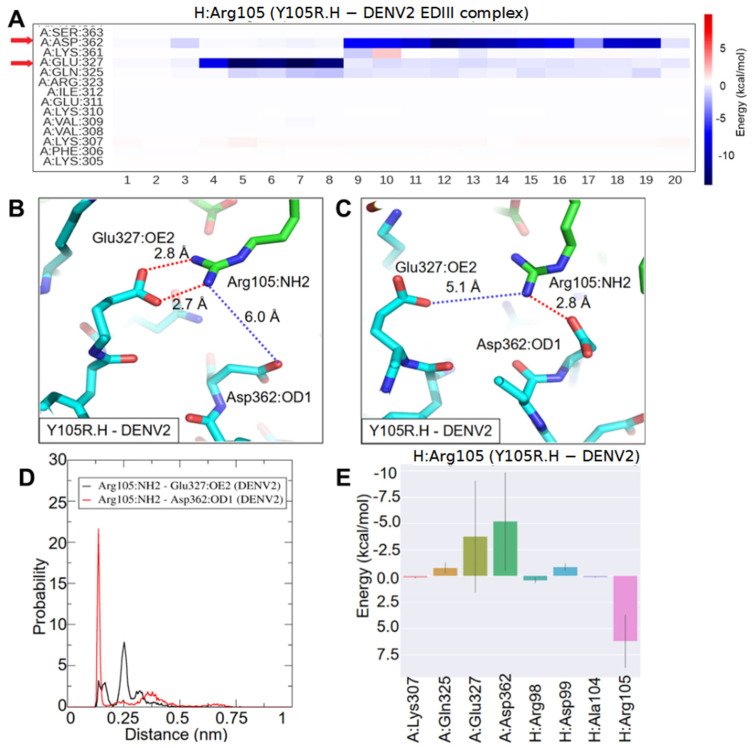
MD simulation of Y105R.H with DENV2 EDIII. (**A**) Energy decomposition analysis: heat map shows energy contribution of residues that interact with Arg105.H per frame in the Y105R.H-DENV2 EDIII complex. Twenty frames with a 2.5-ns interval were extracted for the last 50-ns of the MD simulation. Red arrows highlight residues strongly interacting with Arg105.H. In the Y105R.H-DENV2 EDIII complex, (**B**) new salt bridges were formed between NH2 of Arg105.H and OE2 of Glu327, (**C**) new salt bridges were formed between NH2 of Arg105.H and OD1 of Asp362. Atom pairs are connected by dashed lines in red (interacting) and blue (non-interacting) with distances labeled in representative snapshots. (**D**) Histograms of atom pair distances in the last 50-ns trajectory. Black represents the distance between NH2 of Arg105 and OE2 of Glu327, and red represents the distance between NH2 of Arg105 and OD1 of Asp362. (**E**) Energy contribution of residues that interacted with Arg105.H, corresponding to the heatmap in (**A**). The bars represent averages and solid lines represent the standard deviation. The EDIII, 1A1D-2 heavy chain, and light chain are designated as A, H, and L chain, respectively.

**Figure 4 ijms-23-04197-f004:**
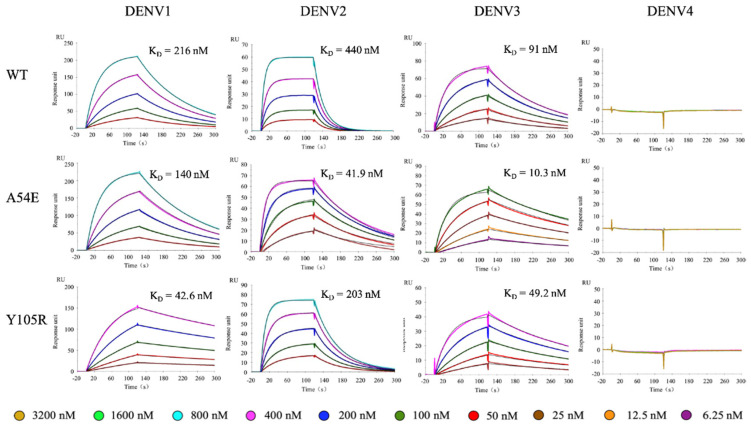
SPR sensorgrams for Fab of 1A1D-2 WT and mutants to DENV1-4. In the SPR experiments, recombinant DENV EDIII was covalently immobilized on the CM5 chip (Cytiva). A series of 1A1D-2 Fab (WT or mutants) in two-fold dilutions were flowed through the chip for two minutes and allowed to dissociated for three minutes. Colored lines represent experiment data at different analyte concentrations, and the black lines are fitting curve. Each experiment was independently repeated for three times. For DENV4, the highest concentration used in this experiment reached 3200 nM for recombinant EDIII proteins. The color keys were provided below the graphs.

**Figure 5 ijms-23-04197-f005:**
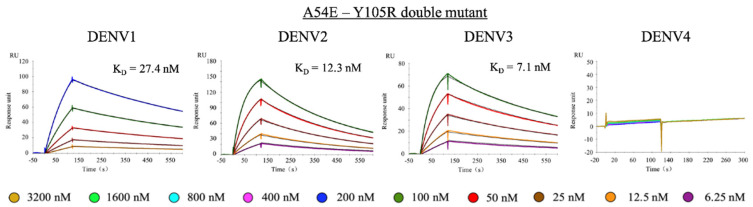
SPR sensorgrams for the double mutant of 1A1D-2 to DENV1, 2 and 3. Colored lines represent experiment data at different analyte concentrations, and the black lines are fitting curve. Each experiment was independently repeated three times. For DENV4, the highest concentration used in this experiment reached 3200 nM for recombinant EDIII proteins. The color keys were provided below the graphs.

**Figure 6 ijms-23-04197-f006:**
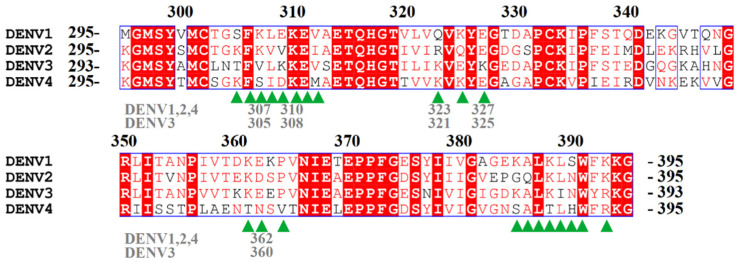
Sequence alignment of DENV1-4 EDIII. DENV3 E protein is two amino acids shorter than other DENV serotypes. Numbering of residues above the sequence is based on DENV1, 2, and 4. The numbers in grey below the sequence highlight key residues mentioned in the main text. The sequence is from DENV1 GuianaFGA891989, DENV2 Thailand/16681/84, DENV3 Philippines/H87/1956, DENV4 Myanmar1976. Epitope residues of 1A1D-2 are indicated using green arrows.

**Figure 7 ijms-23-04197-f007:**
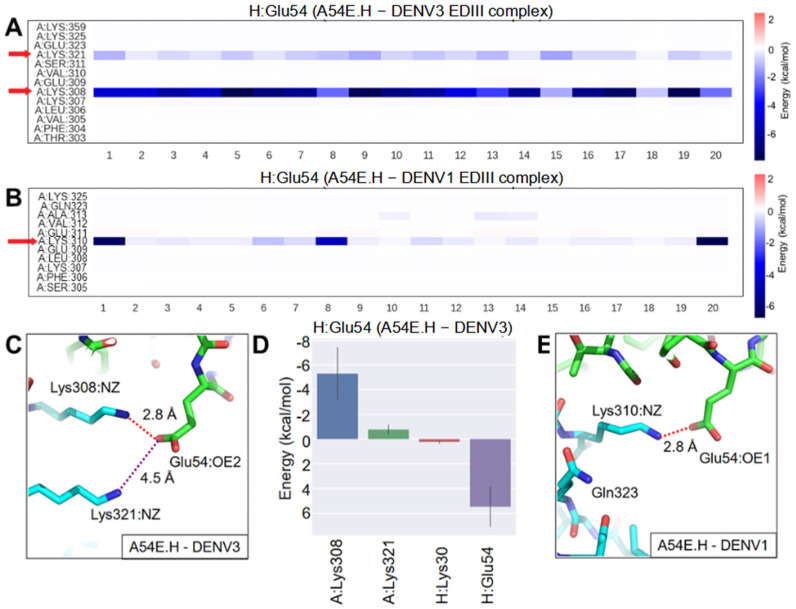
MD simulation of A54E.H with DENV3 and DENV1 EDIII. Energy decomposition analysis: heat maps show energy contribution of residues that interact with Glu54.H per frame in the (**A**) A54E.H-DENV3 EDIII complex and (**B**) A54E.H-DENV1 EDIII complex. Twenty frames with a 2.5-ns interval were extracted for the last 50-ns of the MD simulation. Red arrows highlight residues strongly interacting with Glu54.H. (**C**) In the A54E.H-DENV3 EDIII complex, a stable salt bridge between OE2 of Glu54.H and NZ of Lys308 is highlighted by a red dashed line with the distance labeled in one snapshot. In the same structure, OE2 of Glu54.H and NZ of Lys321 are connected by a purple dashed line with the distance labeled. (**D**) Energy contribution of residues that interacted with Glu54.H, corresponding to the heatmap in (**A**). The bars represent averages and solid lines represent the standard deviation. The EDIII, 1A1D-2 heavy chain and light chain are designated as A, H and L chain, respectively. (**E**) A salt bridge was formed between OE1 of Glu54.H and NZ of Lys310 in the A54E.H-DENV1 complex, highlighted with a red dashed line with the distance labeled.

**Figure 8 ijms-23-04197-f008:**
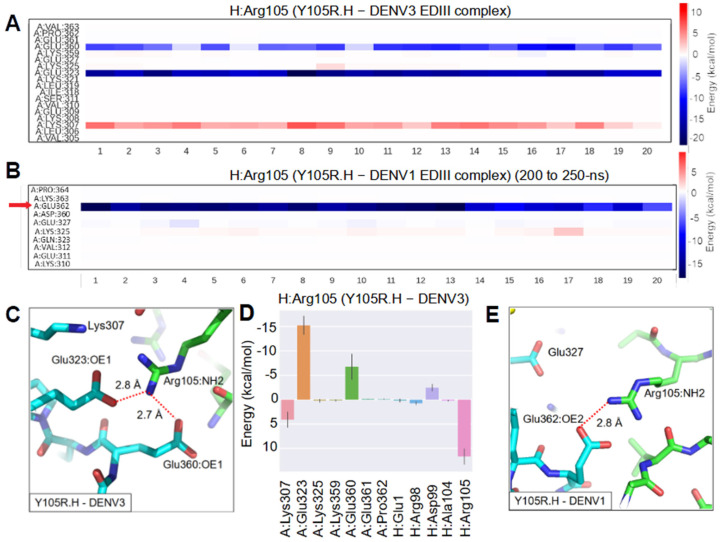
MD simulation of Y105R.H with DENV3 and DENV1 EDIII. Energy decomposition analysis: heat maps show energy contribution of residues that interact with Arg105.H per frame in the (**A**) Y105R.H-DENV3 EDIII complex and (**B**) Y105R.H-DENV1 EDIII complex. Twenty frames with a 2.5-ns interval were extracted from the last 50-ns of the simulation for Y105R.H-DENV3 complex (**A**), and from 200 to 250-ns of the simulation for Y105R.H-DENV1 complex (**B**). Red arrows highlight residues strongly interacting with Arg105.H. (**C**) In the Y105R.H-DENV3 EDIII complex, new salt bridges were formed between (1) NH2 of Arg105.H and OE1 of Glu323 and (2) NH2 of Arg105.H and OE1 of Glu360. The interacting atoms are connected by red dashed lines with distances labeled in a representative snapshot. (**D**) Energy contribution of residues that interacted with Arg105.H, corresponding to the heatmap in (**A**). The bars represent averages and solid lines represent the standard deviation. The EDIII, 1A1D-2 heavy chain and light chain are designated as A, H, and L chain, respectively. (**E**) A salt bridge was formed between NH2 of Arg105.H and OE2 of Glu362 in the Y105R.H-DENV1 complex. The atom pair is connected by a red dashed line with the distance labeled in a representative snapshot.

**Table 1 ijms-23-04197-t001:** Binding kinetics of 1A1D-2 WT and mutants to recombinant EDIII of DENV1, 2, and 3. All experiments were repeated three times.

	k_a_ (1/Ms)	k_d_ (1/s)	K_D_ (nM)	K_D_ Error (nM)
	DENV1			
WT	3.49 × 10^4^	7.55 × 10^−3^	216	7.23
A54E	4.43 × 10^4^	6.17 × 10^−3^	140	4.93
Y105R	4.35 × 10^4^	1.85 × 10^−3^	42.6	1.44
A54E-Y105R	4.39 × 10^4^	1.20 × 10^−3^	27.4	0.51
	DENV2			
WT	1.07 × 10^5^	4.71 × 10^−2^	440	5.51
A54E	1.97 × 10^5^	8.22 × 10^−3^	41.9	2.84
Y105R	9.62 × 10^4^	1.95 × 10^−2^	203	3.21
A54E-Y105R	2.22 × 10^5^	2.74 × 10^−3^	12.3	0.41
	DENV3			
WT	8.70 × 10^4^	7.92 × 10^−3^	91	3.67
A54E	3.54 × 10^5^	3.64 × 10^−3^	10.3	0.52
Y105R	8.52 × 10^4^	4.19 × 10^−3^	49.2	2.87
A54E-Y105R	2.22 × 10^5^	1.58 × 10^−3^	7.11	0.16

## Data Availability

The data presented in this study are available on request from the corresponding author.

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
