# Peer review of "Charged Residue Implantation Improves the Affinity of a Cross-Reactive Dengue Virus Antibody"

_ijms, 2022, doi:10.3390/ijms23084197_

Round 1

Reviewer 1 Report

Review of "Charged residue implantation improves the affinity of a cross-reactive dengue virus antibody"

The authors of this study were interested in examining the ability of the neutralizing monoclonal antibody 1A1D-2 to bind to Dengue EDIII of DENV 1, 2 and 3 serotypes by introducing mutations in the 1A1D-2 heavy chain binding site (A54E and Y105R).  The authors hypothesized that these two amino acid substitutions (based on molecular dynamic simulations) could improve interactions at the binding site and possibly improve the strength of binding.  The effect of the heavy chain mutations on EDIII binding was measured by experimental surface plasmon resonance and computer simulated molecular dynamics analysis. 

Their results indicated that both of these heavy chain mutations increased affinities of 1A1D-2 to DENV 1, 2 & 3 EDIII based on the surface plasmon resonance data in figures 4 & 5 and Table 1.  The molecular dynamics computer simulations indicated that the mutational changes interacted with different amino acids depending on the Dengue serotype.  When the two mutational changes were combined, the affinity of 1A1D-2 to bind EDIII improved by 8 (DENV1), 36 (DENV2) and 13-fold (DENV3).

While the surface plasmon resonance data convincingly demonstrates and supports the author's conclusions, it is unclear to this reviewer whether the mutations in 1A1A-2 will in fact affect the plaque neutralization kinetics towards DENV-1, 2 and 3.  Experimental data will hopefully reflect the predictions made in this manuscript.  

Minor comment- the text in line 184 is completely obscured by the top of Figure 4.  

How realistic is your prediction that 1A1D-2 could be engineered to bind DENV-4 in addition to DENV1, 2 & 3 considering that there was such a variable effect on binding between the 3 serotypes tested?  For example, stronger binding to DENV4 could weaken binding to one or more of the other serotypes.   Please comment in Discussion.

Reviewer 2 Report

In this manuscript, authors reported that charged residue implantation improves the affinity of a cross-reactive dengue virus antibody. All experiments were carefully performed and well conducted to draw the conclusion. However, there are several problems and lacks as following:

  1. In the previous studies, there are many research papers about cross-reactive dengue virus antibodies for the envelope protein domain III (World Intellectual Property Organization, WO2013173348 A1 2013-11-21; Cell Host & Microbe 2010, 8, 271-283; Critical reviews in eukaryotic gene expression 2020, 30, 199-206; Journal of General Virology 2013, 94, 2191-2201; Structure 2018, 26, 51-59.e4; Virology 2009, 392, 103-113; etc.). These reports diminish novelty and significance of the present study. Thus, authors should describe the difference of this study compare with previous reports in detail.
  2. Authors used lots of abbreviation. Please provide the abbreviation list or write full names for the first ones.
  3. For the figure 4, authors have to reduce figure size.
  4. In the figures 4 and 5, please add concentrations for each SPR graph.
  5. After mutations, authors should check affinities for the envelope protein domain III (EDIII) of DENV 4. If done, please even provide negative data.
  6. In the line 256, “the interaction is not very stable” --> “the interaction was quite unstable”
  7. In the line 379, MgCl2 --> MgCl2
  8. Please keep the notation for all references.
